# Associations of genetic and infectious risk factors with coronary heart disease

Flavia Hodel[1,2], Zhi Ming Xu[1,2], Christian Wandall Thorball[3], Roxane de La Harpe[4], Prunelle Letang-Mathieu[1,2], Nicole Brenner[5], Julia Butt[5], Noemi Bender[5], Tim Waterboer[5], Pedro Manuel Marques-Vidal[4], Peter Vollenweider[4], Julien Vaucher[3], Jacques Fellay[1,2,3]*

[1]Global Health Institute, School of Life Sciences, École Polytechnique Fédérale de Lausanne, Lausanne, Switzerland; [2]Swiss Institute of Bioinformatics, Lausanne, Switzerland; [3]Precision Medicine Unit, Lausanne University Hospital and University of Lausanne, Lausanne, Switzerland; [4]Department of Medicine, Internal medicine, Lausanne University Hospital and University of Lausanne, Lausanne, Switzerland; [5]Division of Infections and Cancer Epidemiology, German Cancer Research Center, Heidelberg, Germany

*For correspondence:
jacques.fellay@epfl.ch

**Abstract** Coronary heart disease (CHD) is one of the most pressing health problems of our time and a major cause of preventable death. CHD results from complex interactions between genetic and environmental factors. Using multiplex serological testing for persistent or frequently recurring infections and genome-wide analysis in a prospective population study, we delineate the respective and combined influences of genetic variation, infections, and low-grade inflammation on the risk of incident CHD. Study participants are enrolled in the CoLaus|PsyCoLaus study, a longitudinal, population-based cohort with baseline assessments from 2003 through 2008 and follow-up visits every 5 years. We analyzed a subgroup of 3459 individuals with available genome-wide genotyping data and immunoglobulin G levels for 22 persistent or frequently recurring pathogens. All reported CHD events were evaluated by a panel of specialists. We identified independent associations with incident CHD using univariable and multivariable stepwise Cox proportional hazards regression analyses. Of the 3459 study participants, 210 (6.07%) had at least one CHD event during the 12 years of follow-up. Multivariable stepwise Cox regression analysis, adjusted for known cardiovascular risk factors, socioeconomic status, and statin intake, revealed that high polygenic risk (hazard ratio [HR] 1.31, 95% CI 1.10–1.56, p=2.64 × $10^{-3}$) and infection with *Fusobacterium nucleatum* (HR 1.63, 95% CI 1.08–2.45, p=1.99 × $10^{-2}$) were independently associated with incident CHD. In a prospective, population-based cohort, high polygenic risk and infection with *F. nucleatum* have a small, yet independent impact on CHD risk.

## Editor's evaluation

This study is an important contribution to the understanding of cardiovascular disease aetiology based on solid design and methodology. It is a useful independent replication of the effects of traditional risk factors in a large prospective cohort and a valuable investigation of the role of past infection with a commensal bacterium F. nucleatum which warrants validation.

## Introduction

Worldwide, cardiovascular diseases (CVDs) are the leading cause of mortality (***Roth et al., 2018***). An estimated 17.9 million people die from CVD each year, accounting for 32% of all deaths. CVD is

a broad term for medical conditions involving the heart and blood vessels, such as coronary heart disease (CHD), congenital heart disease, cerebrovascular disease, peripheral arterial disease, rheumatic heart disease, deep vein thrombosis, and pulmonary embolism (*World Health Organization, 2009*).

CHD is the most common type of heart disease (*Roth et al., 2018*). It is caused by atherosclerosis, a build-up of plaque inside the walls of the arteries that supply blood to the heart. CHD progresses over a long period of time and eventually evolves into symptoms such as chest pain (angina), tightness in the chest, breathing difficulties, and pain in the arms or shoulders (*Ambrose and Singh, 2015*). A complete blockage can cause a heart attack.

A combination of demographic, environmental, and genetic factors contribute to the development of CHD (*Khot et al., 2003*; *Mendis et al., 2011*). The main risk factors associated with the development of CHD – smoking, diabetes, hyperlipidemia, and hypertension – have been established by extensive epidemiological research (*MacMahon et al., 1990*; *Stamler et al., 1993*; *Verschuren et al., 1995*; *Weintraub, 1990*). Age is also an important risk factor for CHD (*Castelli, 1984*). Finally, the incidence of CHD is greater in males than in females (*Castelli, 1984*). Very recently, a new algorithm, named Systematic COronary Risk Evaluation 2 (SCORE2), was developed to predict the 10-year risk of first-onset CVD in European populations (*SCORE2 working group and ESC Cardiovascular risk collaboration, 2021*; *SCORE2-OP working group and ESC Cardiovascular risk collaboration, 2021*). This score has replaced the existing HeartScore scoring system, and incorporates most of the risk factors mentioned above (*Conroy et al., 2003*).

CHD also has an important genetic component. In 1938, the first familial risk model for CHD was described and later confirmed by clinical observations and large studies of twins and of longitudinal cohorts (*Müller, 1938*; *Marenberg et al., 1994*; *Samani et al., 2007*; *Abraham et al., 2016*). Based on whole-genome approaches, the heritability of CHD has been estimated at 40–60%, even after controlling for known risk factors (*Vinkhuyzen et al., 2013*).

Multiple clinical studies have identified inflammatory risk factors that are predictive of future cardiovascular events (*Alfaddagh et al., 2020*; *Libby, 2006*; *Hansson, 2005*). Endothelial dysfunction and subintimal cholesterol have been shown to trigger an inflammatory cascade, involving activated macrophages and leading to atherosclerotic lesions. At the molecular level, inflammasome formation in macrophages plays, through their production of interleukin (IL)-1β, an essential role in the propagation of inflammation. These cytokines are released, trigger various inflammatory cells, and produce IL-6 that in turn, stimulate C-reactive protein (CRP) production by the liver, which further enhances the inflammatory cascade within the vascular wall. Today, CRP is an established biomarker of systemic inflammation and a possible predictor of future cardiovascular events (*Libby, 2006*).

The recognition of atherosclerosis as an inflammatory disease has renewed interest in examining the role of pathogens in CHD and other CVDs. Nearly 150 years ago, acute infection with *Bacillus typhosus* was found to cause sclerosing changes in the arterial wall (*Gilbert and Lion, 1889*). A century later, the interest for a potential role of infection in atherosclerosis was renewed, with the discovery that CHD-positive individuals show an increased likelihood of having elevated levels of antibodies to *Chlamydia pneumoniae* (*C. pneumoniae*) (*Saikku et al., 1988*). This was followed by the discovery of the association with CHD of several other infectious agents, including bacteria and viruses, such as *Helicobacter pylori* (*H. pylori*), hepatitis C virus (HCV), and human herpes viruses (*Adinolfi et al., 2018*; *Filardo et al., 2015*; *Wang et al., 2020*; *Zhang et al., 2008*). The exact mechanisms linking infection to low-grade inflammation and atherosclerosis are still being studied, though some potential pathways have been proposed. One proposed mechanism involves the production of pro-inflammatory molecules in response to an infection (*Campbell and Rosenfeld, 2015*). These molecules, such as cytokines, can increase the activity of cells involved in atherosclerosis, such as macrophages and smooth muscle cells, leading to the formation of plaques and other changes in the walls of arteries (*Campbell and Rosenfeld, 2015*). Another mechanism is related to the inflammation at the site of vessel wall. Specifically, it is characterized by the presence of the infectious agents within the atherosclerotic plaques. The infectious consequences on the atherosclerotic plaque can be accelerated progression or a final complication like thrombosis and plaque rupture (*Pedicino et al., 2013*).

Although enormous progress has been made in the understanding of CHD pathogenesis, the overall picture of the combined contribution of infectious, inflammatory, and genetic factors to the risk of developing CHD in the general population remains incomplete. We here use data from the

**eLife** Research article

Genetics and Genomics

CoLaus|PsyCoLaus study, a well-characterized, longitudinal, population-based study from Switzerland, to obtain a more comprehensive view of the evidence for the respective contributions of these factors to CHD.

## Results

### Demographic and serological characteristics

A total of 3459 CoLaus|PsyCoLaus participants with available phenotypic, serological, and genotypic data were included. Their characteristics are presented in *Table 1*.

During the follow-up of 4500 days (12.3 years), at least one CHD event occurred in 210 individuals (6.07%). The number of participants with one, two, and three coronary events was 140, 47, and 14, respectively. Nine individuals had between four and eight coronary events. Eligible study participants were on average 52.8 (standard deviation [SD] ± 10.5) years of age at baseline, 54% were women and 25.5% were smokers. On average, their body mass index (BMI) was 25.5 (±4.3) kg/m$^2$, their systolic blood pressure was 129 (±18) mmHg, and their HDL cholesterol level was 1.66 (±0.43) mmol/L. The percentages of participants by average gross monthly income are 7.5 (<CHF 2999), 19.1 (CHF 3000–4999), 23.3 (CHF 5000–6999), 21.3 (CHF 7000–9499), 14.3 (CHF 9500–13,000), and 14.5% (CHF > 13,000).

For the measured biomarkers of inflammation, the log10-transformed mean (SD) values for high-sensitive CRP (hs-CRP), IL-1β, IL-6, and tumor necrosis factor α (TNF-α) were 0.09 (±0.46), 0.17 (±0.64), 0.24 (±0.58), and 0.46 (±0.38), respectively.

We also investigated participants' serostatus for the following 22 human pathogens: 15 viruses (human polyomaviruses BK [BKV], JC [JCV], 6 [HPyV6], and WU [WUPyV], herpes simplex virus [HSV]-1, HSV-2, varicella zoster virus (VZV), Epstein–Barr virus [EBV], cytomegalovirus [CMV], human herpes virus 6A [HHV-6A], HHV-6B, HHV-7, Kaposi's sarcoma-associated herpes virus [KSHV], parvovirus B19 [PVB-19], and rubella virus); six bacteria (*Chlamydia trachomatis* [*C. trachomatis*], *Clostridium tetani* [*C. tetani*], *Cornybacterium diphteriae* [*C. diphteriae*], *F. nucleatum*, *H. pylori*, and *S. gallolyticus*); and one parasite (*T. gondii*) (*Appendix 1—table 1*). The overall seropositivity ranged from 3.99% (*S. gallolyticus*) to 96.80% (EBV). The overall serostatus split between CHD-positive (with at least one CHD event during follow-up) and CHD-negative individuals are shown in *Figure 1*. Rubella, *C. tetani* and *C. diphteriae* were excluded from further analyses as the antibodies detected against these pathogens were most likely induced by vaccination.

### Univariable predictors of CHD incidence

To validate the utility of SCORE2 in our cohort, we tested its association with CHD. SCORE2 was significantly and positively associated with CHD (HR 1.72, 95% CI 1.61–1.85, p = 2.87×10$^{-61}$) (*Appendix 1—table 2*). We also observed a significant inverse association between average gross monthly income and CHD risk (HR 0.85, 95% CI 0.76–0.96, p = 7.27×10$^{-3}$).

To investigate the relationship between CHD and humoral response to infectious agents, we tested the association of serostatus for each of the included 19 persistent or frequently occurring pathogens with CHD. We found significant positive associations for six of them, including three herpes viruses, namely HSV-1 (HR 1.88, 95% CI 1.30–2.68, p = 6.52×10$^{-4}$), HHV-6A (HR 1.39, 95% CI 1.03–1.86, p = 2.89×10$^{-2}$), and VZV (HR 1.70, 95% CI 1.02–2.82, p = 4.25×10$^{-2}$), one polyomavirus, HPyV6 (HR 1.66, 95% CI 1.06–2.61, p = 2.74×10$^{-2}$), and two bacteria, *F. nucleatum* (HR 1.66, 95% CI 1.20–2.29, p = 2.32×10$^{-3}$), and *C. trachomatis* (HR 1.45, 95% CI 1.11–1.91, p = 7.22×10$^{-3}$) (*Appendix 1—table 2*).

To evaluate the impact of the biomarkers of inflammation on CHD risk, we tested the association of log10-transformed levels of hs-CRP, IL-1β, IL-6, and TNF-α with CHD. We observed a positive relationship between individuals' hs-CRP (HR 1.91, 95% CI 1.42–2.55, p = 1.51×10$^{-5}$) and TNF-α (HR 1.43, 95% CI 1.05–1.96, p = 2.46×10$^{-2}$) levels, and increased risk of CHD event (*Appendix 1—table 2*).

Finally, we calculated a CHD polygenic risk score (CHD-PRS) for each subject to investigate the effect of common human genetic variations on CHD. As expected, we observed a significant association between the PRS and CHD (HR 1.32, 95% CI 1.16–1.51, p = 4.29×10$^{-5}$), confirming that genetic predisposition to CHD can be captured through CHD-PRS. The top three genetic principal components (PC1, PC2, and PC3) were not significantly associated with CHD (*Appendix 1—table 2*).

**Table 1.** Baseline characteristics of 3459 CoLaus|PsyCoLaus participants by coronary heart disease (CHD) cases and controls.

p-Values are based on the t-test for continuous variables and Fisher's exact test for categorical variables comparing the CHD cases and controls group.

| | Overall N = 3249 (100%) | Controls N = 3249 (93.93%) | CHD cases N = 210 (6.07%) | p |
|---|---|---|---|---|
| **Baseline characteristics** | | | | |
| Age (mean [SD]) | 52.83 [10.48] | 52.34 [10.36] | 60.34 [9.53] | <0.001 |
| BMI (mean [SD]) | 25.51 [4.31] | 25.41 [4.27] | 27.11 [4.62] | <0.001 |
| Systolic blood pressure (mean [SD]) | 129.04 [18.40] | 128.32 [18.01] | 140.18 [20.60] | <0.001 |
| HDL cholesterol (mean [SD]) | 1.66 [0.43] | 1.67 [0.43] | 1.49 [0.41] | <0.001 |
| LDL cholesterol (mean [SD]) | 3.34 [0.92] | 3.33 [0.92] | 3.48 [0.91] | 0.018 |
| Total cholesterol (mean [SD]) | 5.60 [1.03] | 5.59 [1.02] | 5.73 [1.04] | 0.051 |
| Sex = male (%) | 1592 (46.0) | 1448 (44.6) | 144 (68.6) | <0.001 |
| Statin = yes (%) | 296 (8.6) | 242 (7.4) | 54 (25.7) | <0.001 |
| Average gross monthly income (in CHF): | | | | 0.029 |
| <2999 (%) | 178 (5.1) | 161 (5.0) | 17 (8.1) | |
| 3000–4999 (%) | 452 (13.1) | 425 (13.1) | 27 (12.9) | |
| 5000–6999 (%) | 552 (16.0) | 517 (15.9) | 35 (16.7) | |
| 7000–9499 (%) | 504 (14.6) | 468 (14.4) | 36 (17.1) | |
| 9500–13,000 (%) | 338 (9.8) | 323 (9.9) | 15 (7.1) | |
| >13,000 (%) | 344 (9.9) | 335 (10.3) | 9 (4.3) | |
| Refused or missing (%) | 1091 (31.5) | 1020 (31.4) | 71 (33.8) | |
| Smoking = yes (%) | 883 (25.5) | 820 (25.2) | 63 (30.0) | 0.146 |
| Genetics | | | | |
| CHD-PRS (mean [SD]) | 0.00 [1.00] | −0.02 [0.99] | 0.26 [1.01] | <0.001 |
| Biomarkers of inflammation | | | | |
| hs-CRP (mean [SD]) | 0.09 [0.46] | 0.08 [0.46] | 0.22 [0.44] | <0.001 |
| TNF-α (mean [SD]) – 63 NAs | 0.46 [0.38] | 0.46 [0.38] | 0.53 [0.35] | 0.013 |
| IL-1β (mean [SD]) – 1'319 NAs | 0.17 [0.64] | 0.17 [0.64] | 0.14 [0.67] | 0.637 |
| IL-6 (mean [SD]) – 294 NAs | 0.24 [0.58] | 0.24 [0.58] | 0.28 [0.55] | 0.398 |
| Persistent pathogens | | | | |
| Human polyomaviruses: | | | | |
| BKPyV = seropositive (%) | 2912 (84.2) | 2735 (84.2) | 177 (84.3) | 1.000 |
| JCPyV = seropositive (%) | 1812 (52.4) | 1696 (52.2) | 116 (55.2) | 0.434 |
| HPyV6 = seropositive (%) | 2948 (85.2) | 2759 (84.9) | 189 (90.0) | 0.056 |
| WUPyV = seropositive (%) | 3309 (95.7) | 3105 (95.6) | 204 (97.1) | 0.362 |
| Human herpes viruses: | | | | |
| HSV-1 = seropositive (%) | 2547 (73.6) | 2373 (73.0) | 174 (82.9) | 0.002 |
| HSV-2 = seropositive (%) | 601 (17.4) | 564 (17.4) | 37 (17.6) | 0.998 |
| CMV = seropositive (%) | 1868 (54.0) | 1756 (54.0) | 112 (53.3) | 0.897 |
| EBV = seropositive (%) | 3350 (96.8) | 3147 (96.9) | 203 (96.7) | 1.000 |

*Table 1 continued on next page*

*Table 1 continued*

| | Overall<br>N = 3249 (100%) | Controls<br>N = 3249 (93.93%) | CHD cases<br>N = 210 (6.07%) | p |
|---|---|---|---|---|
| **Baseline characteristics** | | | | |
| HHV-6A = seropositive (%) | 865 (25.0) | 800 (24.6) | 65 (31.0) | **0.049** |
| HHV-6B = seropositive (%) | 1373 (39.7) | 1295 (39.9) | 78 (37.1) | 0.480 |
| HHV-7 = seropositive (%) | 1846 (53.4) | 1746 (53.7) | 100 (47.6) | 0.099 |
| KSHV = seropositive (%) | 141 (4.1) | 133 (4.1) | 8 (3.8) | 0.983 |
| VZV = seropositive (%) | 3047 (88.1) | 2853 (87.8) | 194 (92.4) | 0.061 |
| Parvovirus: | | | | |
| PVB-19 = seropositive (%) | 2420 (70.0) | 2277 (70.1) | 143 (68.1) | 0.595 |
| Bacteria: | | | | |
| *C. trachomatis* = seropositive (%) | 1213 (35.1) | 1120 (34.5) | 93 (44.3) | **0.005** |
| *F. nucleatum* = seropositive (%) | 520 (15.0) | 473 (14.6) | 47 (22.4) | **0.003** |
| *H. pylori* = seropositive (%) | 685 (19.8) | 645 (19.9) | 40 (19.0) | 0.846 |
| *S. gallolyticus* = seropositive (%) | 135 (3.9) | 130 (4.0) | 5 (2.4) | 0.322 |
| Parasite: | | | | |
| *T. gondii* = seropositive (%) | 1445 (41.8) | 1349 (41.5) | 96 (45.7) | 0.262 |

## Co-linearity and proportional hazard assumption testing

We calculated pairwise correlations between all variables that were found to be significant in univariable analyses. *Appendix 2—figure 1* and *Appendix 2—figure 2* illustrate that no strong correlations exist between significant variables. The strongest correlation was observed between SCORE2 and hs-CRP, and between seropositivity to *C. trachomatis* and gross monthly household income, with Pearson's and Cramer's V coefficients of 0.22 and 0.15, respectively. The proportionality assumption was tested for all significant variables using the Schoenfeld residuals. The residual tests indicated that all variables satisfied the proportional hazards assumption, revealing that the effect of all covariates are constant in time (*Appendix 2—figure 3*). Finally, we also assessed potential co-linearity issues among predictors that could affect model fitting. No variance inflation factor (VIF) value was indicative of co-linearity.

## Multivariable model

To identify the independent risk factors of CHD in our cohort, we performed backward stepwise selection on 2323 individuals with non-missing data using a multivariable Cox proportional hazards model, starting with all the significant factors from the univariable models. The final multivariable analysis confirmed that SCORE2 (HR 1.96 per SD increase, 95% CI 1.74–2.22, p = $2.42 \times 10^{-27}$) is an independent prognostic factor of CHD (*Figure 2*). We also observed significant independent associations for statin intake (HR 2.24, 95% CI 1.50–3.35, p = $9.17 \times 10^{-5}$) and for seropositivity to *F. nucleatum* infection (HR 1.63, 95% CI 1.08–2.45, p = $1.99 \times 10^{-2}$). Comparing individuals who had a least one CHD event (CHD group) against those who had no event during the follow-up period (control group), 22.4% (47/210) of the individuals in the CHD group were seropositive to *F. nucleatum*, versus 14.6% (473/3249) in the control group (p = 0.003) (*Figure 1*, *Table 1*). Lastly, we also observed a significant association between CHD occurrence and elevated CHD-PRS with an HR of 1.31 (95% CI 1.10–1.56, p = $3.32 \times 10^{-3}$) per SD increase.

To assess if the overall burden of infections contributed to increased risk of CHD, study participants were stratified according to their overall seropositivity index for measured pathogens, calculated by summing the number of pathogens for which they show seropositivity (range: 0–16). The numbers of individuals in each pathogen burden stratum are shown in *Appendix 2—figure 4*. In the univariable Cox model, pathogen burden significantly increased the risk of CHD occurrence (HR 1.11, 95% CI

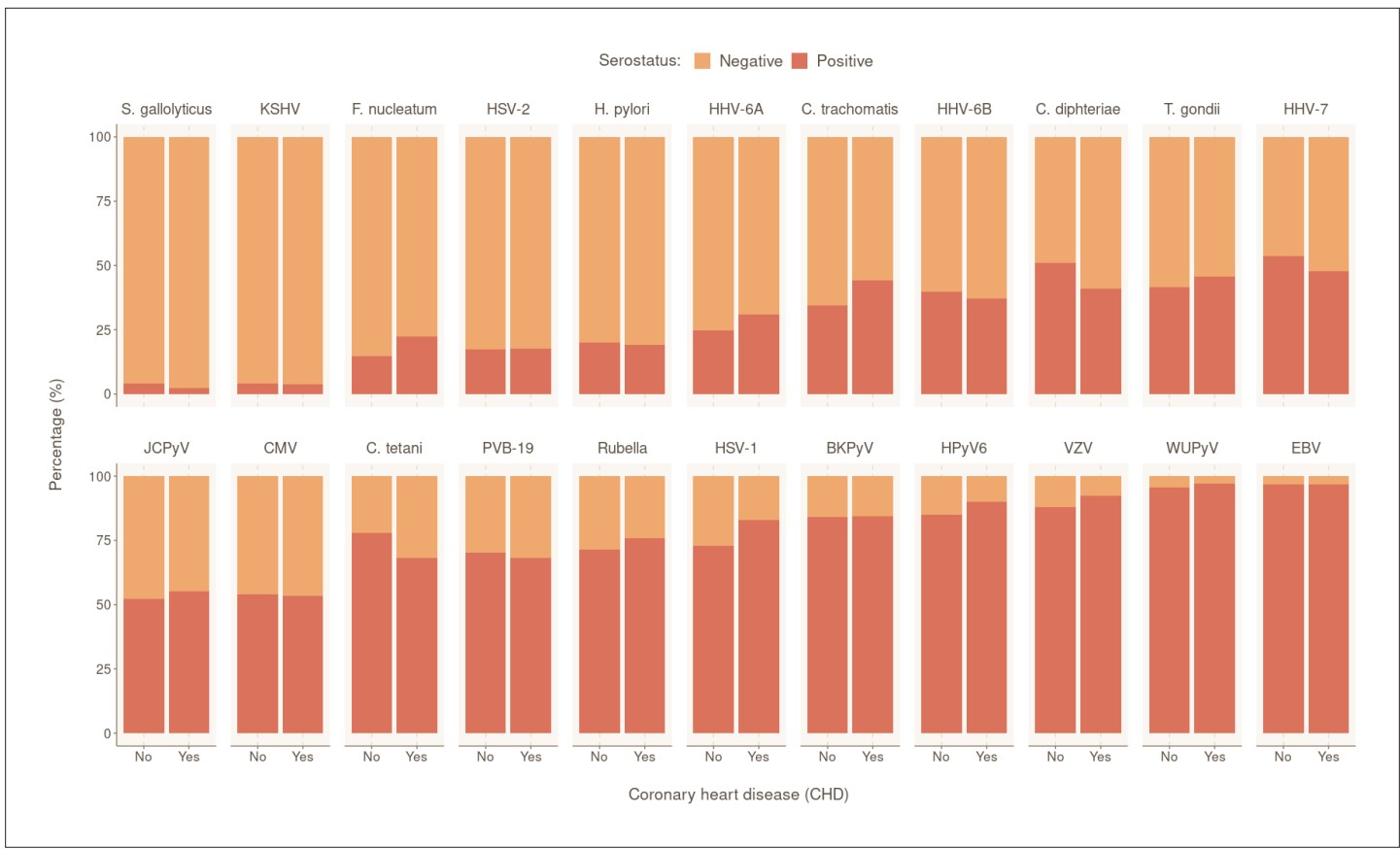

**Figure 1.** Prevalence of tested pathogens in CoLaus|PsyCoLaus study in participants with and without coronary heart disease (CHD). Overall serostatus for the 22 pathogens are shown in the CHD-positive group (individuals with at least one CHD event during follow-up) or CHD-negative group. The y-axis indicates the relative percentage within each group. Pathogens are ranked in ascending order of overall seropositivity (all individuals combined).

The online version of this article includes the following source data for figure 1:

**Source data 1.** Data underlying *Figure 1*.

1.03–1.18, p = 3.25×10⁻³) (*Appendix 1—table 2*). However, after adjustment with multivariable Cox proportional hazards regression, pathogen burden did not meet the level of significance for staying in the model.

## Discussion

CHD is a complex disease that is influenced by demographic, environmental, and genetic factors (*Khot et al., 2003*; *Mendis et al., 2011*). Infections have also been suspected to increase the risk of CHD, directly or through the induction of chronic inflammation (*Vojdani, 2003*). The present study investigated the independent and combined effects of these risk factors as possible prognostic indicators for the occurrence of CHD. We performed an event-free survival analysis of incident CHD using data from a longitudinal, population-based study, in which more than 6% of participants developed CHD over a 12-year study period.

We confirmed the utility of SCORE2 to predict CHD risk in our cohort (*SCORE2 working group and ESC Cardiovascular risk collaboration, 2021*). Of note, chronic inflammation reflected in hs-CRP level did not appear as an independent predictor of CHD in our analyses, as the univariable association signal was suppressed after adjustment for SCORE2 levels.

We studied the effect of human genetic determinants on CHD occurrence using PRS, and we reproduced previously observed effects: participants with a higher CHD-PRS have a greater risk of CHD, even after adjustment for all known factors (*Ding et al., 2011*; *Kullo et al., 2016*). This result confirms the existence of genetic susceptibility loci for CHD, and that the individual genetic

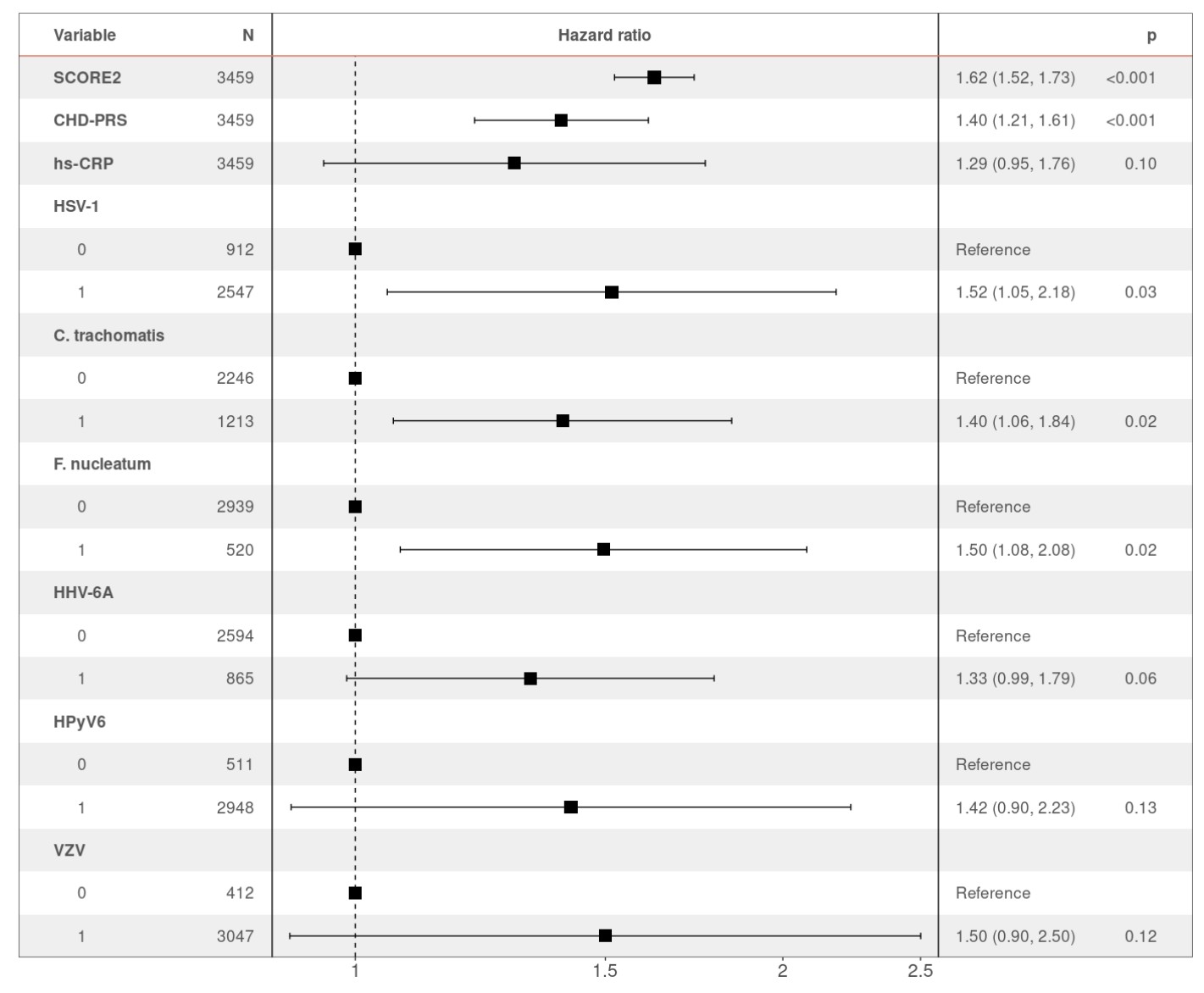

**Figure 2.** Hazard ratio (HR) and 95% confidence intervals of coronary heart disease (CHD) occurrence according to associated factors. HR > 1 indicates an increased risk of CHD, whereas HR < 1 indicates a protective effect. p-Values (**p**) for each factor based on the multivariable Cox regression are shown.

The online version of this article includes the following source data for figure 2:

**Source data 1.** Data underlying *Figure 2*.

background modulates CHD risk independently from age, sex, or co-morbidities. Our work confirms the potential interest in using PRS to improve the prediction of coronary events.

We also evaluated the potential contribution of multiple persistent or frequently recurring pathogens to CHD after controlling for conventional CHD risk factors, socioeconomic status, and human genetic variability. We observed an association of CHD with detection of antibodies against *F. nucleatum*. This pathogen is very prevalent in humans (*Adams et al., 2004*; *Afra et al., 2013*; *Looker et al., 2015*). *F. nucleatum* is an anaerobic bacterium that belongs to the normal flora of the oral cavity and plays an important role in the development and progression of oral diseases, such as gingivitis (gum inflammation) and periodontitis (infection of the gums). Under pathological conditions, the pathogen can spread by the hematogenous route to extra-oral systemic sites, including the gut and the female genital tract (*Han and Wang, 2013*; *Han et al., 2004*). Studies have also suggested the involvement of *F. nucleatum* in CVD. First, by its capacity to directly migrate into arterial plaques, thus

exacerbating atherosclerosis, and more recently, through the association of periodontitis and CVD (*Kholy et al., 2015*; *Elkaïm et al., 2008*; *Figuero et al., 2011*; *Ford et al., 2006*; *Han, 2015*; *Zardawi et al., 2020*). Finally, it has been shown that periodontal pathogens are able to spread through the bloodstream from the buccal cavity to the arteries in patients with detectable coronary calcium, a very specific marker of atherosclerosis (*Corredor et al., 2022*). In summary, the relationship between oral inflammations and CVD could be explained by the colonization of arterial walls and atherosclerosis plaques by dental bacteria, as well as by increased systemic inflammation due to oral infection. However, to date, no direct causality has been established. Besides, no genome-wide association study on *F. nucleatum* has been published, neither on humoral immune response (i.e., IgG levels) nor on susceptibility to infection/colonization (i.e., serostatus).

HSV-1, HHV-6A, VZV, HPyV6, and *C. trachomatis* serologies, as well as total burden of infection, were associated with CHD occurrence in univariable models. However, these factors were not significantly associated in the multivariable analysis, suggesting that at least some of them could be indirect markers of socioeconomic status.

Our data do not support the existence of the previously identified associations between CHD and *H. pylori*, or CMV. The conflicting reports of possible associations between these pathogens and CHD could be due to sample size but remain questionable. Further extensive, and high-quality studies are needed to thoroughly examine these associations and provide firm conclusions.

Our study has several limitations. As is the case for most longitudinal studies, the absence of data on individuals who dropped out before the end of the follow-up implies that some CHD events could have gone undetected. Also, the demographic information, as well as the clinical and laboratory measurements, were obtained at baseline, and we do not know whether participant information changed over time. Adjustment for risk factors measured at baseline does not account for clinical or demographic changes that could influence CHD outcomes. Similarly, we do not know how the antibody responses against the various antigens evolved over the 12 years of the study. In addition, no significance adjustment was performed when using multiple univariable tests to determine the effect of single factors on CHD risk, although this may increase the false positive rate. Moreover, we were unable to replicate previously published observations of associations of CHD with *C. pneumoniae* and HCV as serologies for these pathogens were not available. From a more practical point of view, the identified association with *F. nucleatum* needs to be replicated and validated in independent cohorts and different populations. Finally, the clinical utility of including genetic and infection biomarkers in CHD prediction algorithms will need to be demonstrated.

## Conclusion

CHD is a multicomponent disease that is caused by demographic, environmental, and genetic factors. Inflammation, possibly caused by persistent or frequently recurring infections, can contribute to its development. We identified a statistically significant association between the incidence of CHD and *F. nucleatum* infection, after adjustment for all established risk factors. We also confirmed that the individual polygenic risk of CVD, calculated from genome-wide genotypes, represents an independent risk factor for incident CHD. Our results can help to better identify subjects at high risk for CHD and provide a rationale for future anti-infective prevention trials.

## Methods
### Study cohort

The CoLaus|PsyCoLaus study is a longitudinal population-based study initiated in Lausanne in 2003; it mainly investigates the biological, environmental, and genetic determinants of CVD (https://www.colaus-psycolaus.ch/) (*Firmann et al., 2008*). The study involves over 6500 participants of European ancestry, who were recruited at random from the general population and represent approximately 10% sample of Lausanne citizens. Of the participants, 47.5% are men, and age at enrolment ranged from 35 to 75 years (mean ± SD: 51 ± 10.9). The study participants provided detailed phenotypic information through questionnaires, interviews, clinical and biological data. Nuclear deoxyribonucleic acid (DNA) was also extracted from the blood for whole-genome genotyping data. Every 5 years, follow-up interviews on the participants' lifestyle and health status are conducted. There are three completed follow-ups and a fourth follow-up began in January 2022. The institutional Ethics

Committee of the University of Lausanne, which later became the Ethics Commission of the Canton Vaud (https://www.cer-vd.ch/), approved the CoLaus|PsyCoLaus study (reference 16/03, decisions of January 13 and February 10, 2003), and all participants gave written consent.

### Cardiovascular phenotype

The medical records of the participants who reported a CHD event during their lifetime were collected and evaluated by an independent panel of specialists. Information on the cause of death was also collected prospectively during the study period. The full procedure was described previously (*Beuret et al., 2021*). Only first events occurring after the baseline and up to day 4500 after the baseline were included in the analysis, as only during this period were all participants reliably followed.

### DNA genotyping data and PRS calculation for cardiovascular phenotypes

The BB2 GSK-customized Affymetrix Axiom Biobank array was used to genotype DNA samples from 5399 participants at approximately 800,000 single nucleotide polymorphisms (SNPs). After genotype imputation and quality control procedures, approximately 9 million SNPs were available for analysis (*Hodel et al., 2021*). We then calculated, based on the risk effects of common SNPs, the CHD-PRS for each study participant. We used validated PRS from Inouye et al., available in the polygenic score catalog (*Inouye et al., 2018*; *Lambert et al., 2021*). These scores and summary statistics were used to construct the CHD-PRS in our target cohort data by using the clumping and thresholding method of the PRSice-2 v2.2.7 software (*Choi et al., 2020*). A standardized method was used to obtain the PRS, by multiplying the risk allele dosage for each variant by the effect size and summing the scores across all selected variants. SNPs were clumped according to linkage disequilibrium (r2 < 0.1) within a 250 kb window.

### CHD risk evaluation

The risk of CHD for each participant was also assessed using the very recent SCORE2 and SCORE2-Older Persons (SCORE2-OP, for individuals >65 years of age) algorithms (*SCORE2-OP working group and ESC Cardiovascular risk collaboration, 2021*; *SCORE2 working group and ESC Cardiovascular risk collaboration, 2021*). These two algorithms will be referred to as SCORE2. SCORE2 was derived, calibrated, and validated to predict the 10-year risk of first-onset CVD using data from 13 million individuals from >50 European prospective studies and national registries. To develop this algorithm, the authors used competing risk-adjusted and age- and sex-specific models including age, current smoking, systolic blood pressure, and total, low-density lipoprotein (LDL), and high-density lipoprotein (HDL) cholesterol. The authors also defined four risk regions in Europe on the basis of country-specific CVD mortality. For CoLaus|PsyCoLaus participants, calculations were based on the low-risk region corresponding to Switzerland. The raw scores of participants were standardized to Z-scores with approximately zero mean and unit variance before data analysis.

### Measurement of inflammatory biomarkers

Venous blood samples (50 mL) of the participants, in a fasted state, were drawn. Before cytokine assessment, the serum blood samples were stored at −80°C, then they were sent to the laboratory on dry ice. The measurements of hs-CRP, IL-1β, IL-6, and TNF-α cytokine levels were described previously in detail (*Marques-Vidal et al., 2011*). Briefly, hs-CRP levels were assessed by immunoassay and latex HS (IMMULITE 1000-High, Diagnostic Products Corporation, Los Angeles, CA, USA). Cytokine levels were measured using a multiplexed particle-based flow cytometric cytokine assay on the flow cytometer (FC500 MPL, BeckmanCoulter, Nyon, Switzerland), thus following the manufacturer's instructions. The lower limits of detection for IL-1β, IL-6, and TNF-α were 0.2 pg/mL. Intra- and inter-assay coefficients of variation were, respectively, 15% and 16.7% for IL-1β, 16.9% and 16.1% for IL-6, and 12.5% and 13.5% for TNF-α. For quality control, repeat measurements were performed on 80 subjects randomly selected from the initial sample. Individuals with hs-CRP levels above 20 mg/L were assigned a value of 20 by the manufacturer therefore were removed from the hs-CRP analyses as indicative of acute inflammation.

### Serological analyses

To assess the humoral responses to a total of 38 antigens derived from 22 persistent infectious agents, serum samples were analyzed by the Infections and Cancer Epidemiology Division at the German

Cancer Research Center (Deutsches Krebsforschungszentrum [DKFZ]) in Heidelberg (*Waterboer et al., 2005*; *Waterboer et al., 2006*). Studied pathogens included 15 viruses (BKV, JCV, HPyV6, WUPyV, HSV-1, HSV-2, VZV, EBV, CMV, HHV-6A, HHV-6B, HHV-7, KSHV, PVB-19, and rubella virus); six bacteria (*C. diphteriae*, *C. tetani*, *C. trachomatis*, *F. nucleatum*, *H. pylori*, and *S. gallolyticus*); and one parasite (*T. gondii*) (for details, see *Appendix 1—table 1*). The seroreactivity was measured at a serum dilution of 1:1000 by using multiplex serology based on glutathione S-transferase fusion capture immunosorbent assays combined with fluorescent bead technology. For each infectious agent tested, the antibody responses were measured for one to six antigens and then expressed as a binary result (IgG positive or negative), based on the predefined median fluorescence intensity thresholds. To define overall seropositivity against infectious agents when more than one antigen was used, we applied the pathogen-specific algorithms suggested by the manufacturer (see references in *Appendix 1—table 1*).

## Statistical analyses

Univariable and multivariable Cox proportional hazard models were used to explore the relationship between risk factors and CHD incidence in the CoLaus|PsyCoLaus study. Each variable was first screened in the univariable model. To identify potential confounding due to population structure, we also tested the top three genetic principal components (PC1, PC2, and PC3) for association with CHD. We then examined the proportional hazards assumption of the significant ($p < 0.05$) covariates by using the scaled Schoenfeld residuals. The residuals were plotted over time for each covariate to test for time independence. Risk factors significantly associated with CHD in the univariable model were further evaluated using pairwise correlations. Finally, the identified risk factors were assessed using multivariable stepwise Cox regression analysis, adjusted for competing risk (i.e., SCORE2), socio-economic status (i.e., gross monthly household income), and statin intake. Potential multicollinearity between statistically significant factors ($p < 0.05$) were identified using VIFs. The existence of multicollinearity between co-variates was determined by a VIF value > 2. We performed all statistical analyses using R (version 4.2.1).

## Acknowledgements

We thank the participants in the CoLaus|PsyCoLaus study for their time and contribution to this study. We also thank all the clinical, academic, and administrative collaborators who helped with participant recruitment, study coordination, data collection, and storage. This project was supported by the Swiss National Science Foundation (grant 31003A_175603 to JF). The CoLaus|PsyCoLaus study was and is supported by research grants from GlaxoSmithKline, the Faculty of Biology and Medicine of Lausanne, and the Swiss National Science Foundation (grants 3200B0_105993, 3200B0_118308, 33CSCO_122661, 33CS30_139468, 33CS30_148401, and 33CS30_177535/1).

## Additional information

### Funding

| Funder | Grant reference number | Author |
| --- | --- | --- |
| Schweizerischer Nationalfonds zur Förderung der Wissenschaftlichen Forschung | 31003A_175603 | Jacques Fellay |

The funders had no role in study design, data collection and interpretation, or the decision to submit the work for publication.

### Author contributions

Flavia Hodel, Conceptualization, Software, Formal analysis, Visualization, Methodology, Writing – original draft; Zhi Ming Xu, Conceptualization, Methodology; Christian Wandall Thorball, Julien Vaucher, Resources, Methodology; Roxane de La Harpe, Nicole Brenner, Julia Butt, Noemi Bender,

Resources; Prunelle Letang-Mathieu, Formal analysis; Tim Waterboer, Resources, Investigation; Pedro Manuel Marques-Vidal, Resources, Data curation, Investigation; Peter Vollenweider, Investigation; Jacques Fellay, Supervision, Funding acquisition, Writing – original draft, Project administration

**Author ORCIDs**
Flavia Hodel (iD) http://orcid.org/0000-0001-7331-7357
Nicole Brenner (iD) http://orcid.org/0000-0002-7690-4925
Noemi Bender (iD) http://orcid.org/0000-0003-2542-9949
Pedro Manuel Marques-Vidal (iD) http://orcid.org/0000-0002-4548-8500
Jacques Fellay (iD) http://orcid.org/0000-0002-8240-939X

**Ethics**
Human subjects: The institutional Ethics Committee of the University of Lausanne, which later became the Ethics Commission of the Canton Vaud (www.cer-vd.ch), approved the CoLaus|PsyCoLaus study (reference 16/03, decisions of 13th January and 10th February 2003), and all participants gave written consent.

**Decision letter and Author response**
Decision letter https://doi.org/10.7554/eLife.79742.sa1
Author response https://doi.org/10.7554/eLife.79742.sa2

---

## Additional files

**Supplementary files**
• MDAR checklist

**Data availability**
The data of CoLaus|PsyCoLaus study used in this article cannot be fully shared as they contain potentially sensitive personal information on participants. According to the Ethics Committee for Research of the Canton of Vaud, sharing these data would be a violation of the Swiss legislation with respect to privacy protection. However, coded individual-level data that do not allow researchers to identify participants are available upon request to researchers who meet the criteria for data sharing of the CoLaus|PsyCoLaus Datacenter (CHUV, Lausanne, Switzerland). Any researcher affiliated to a public or private research institution who complies with the CoLaus|PsyCoLaus standards can submit a research application to research.colaus@chuv.ch or research.psycolaus@chuv.ch. Proposals requiring baseline data only, will be evaluated by the baseline (local) Scientific Committee (SC) of the CoLaus and PsyCoLaus studies. Proposals requiring follow-up data will be evaluated by the follow-up (multicentric) SC of the CoLaus|PsyCoLaus cohort study. Detailed instructions for gaining access to the CoLaus|PsyCoLaus data used in this study are available at https://www.colaus-psycolaus.ch/professionals/how-to-collaborate/. The underlying code used to analyze the data in this manuscript is publicly available on GitHub (https://github.com/flaviahodel/cox-chd-analysis, copy archived at swh:1:rev:317cf14e0fbf09f214b-792cc5ea5a399739e15a1). Source Data files have been provided for all Figures and Figure Supplements. "Figure 1 - Source Data", "Figure 2 - Source Data", "Figure S1 - Source Data", "Figure S2 - Source Data", "Figure S3 - Source Data", and "Figure S4 - Source Data" contain the numerical data used to generate the figures.

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

# Appendix 1

## Additional tables

**Appendix 1—table 1.** Characteristics of infectious agent-specific antigens used on the Multiplex Serology platform in CoLaus|PsyCoLaus.

| Family | Pathogen | Antigen | (Predicted) function | Def. of seropositivity is based on | Reference |
|---|---|---|---|---|---|
| Human polyomaviruses | BKV | VP1 | Major capsid protein | NA | *Kjaerheim et al., 2007*; *Gossai et al., 2016*; *Robles et al., 2015* |
| | JCV | VP1 | Major capsid protein | NA | *Kjaerheim et al., 2007*; *Gossai et al., 2016*; *Robles et al., 2015* |
| | HPyV6 | VP1 | Major capsid protein | NA | *Kjaerheim et al., 2007*; *Gossai et al., 2016*; *Robles et al., 2015* |
| | WUPyV | VP1 | Major capsid protein | NA | *Kjaerheim et al., 2007*; *Gossai et al., 2016*; *Robles et al., 2015* |
| Human herpes viruses | CMV | pp150 | Tegument protein | At least two positive | *Brenner et al., 2018* |
| | | pp52 | DNA binding protein | | |
| | | pp28 | Capsid protein | | |
| | EBV | ZEBRA | Replication activator | At least two positive | *Brenner et al., 2018* |
| | | EA-D | Replication (polymerase accessory subunit) | | |
| | | VCA p18 | Capsid protein | | |
| | | EBNA1 | Replication, latent viral infection | | |
| | HHV-6 | IE1B | Potential transactivator | Any HHV-6=at least one positive HHV-6A=IE1A and/or p100HHV-6B=IE1B and/or p101K | *Bassig et al., 2018*; *Engdahl et al., 2019*; *Freuer et al., 2020* |
| | | IE1A | Potential transactivator | | |
| | | p101K | Potential tegument protein | | |
| | | p100 | Potential tegument protein | | |
| | HHV-7 | U14 | Potential tegument protein | NA | Validation ongoing |
| | HSV-1 | gG | Membrane glycoprotein | NA | *Brenner et al., 2019a* |
| | HSV-2 | mgG | Membrane glycoprotein | NA | *Brenner et al., 2019a* |
| | KSHV | LANA3 | Replication and long-term persistence | At least one positive | Validation ongoing |
| | | K8.1 | Structural glycoprotein | | |
| | VZV | gE/gI | Envelope glycoprotein | NA | *Brenner et al., 2019a* |
| Parvovirus | B19 | VP1unique | Minor capsid protein | NA | *Brenner et al., 2019b* |

*Appendix 1—table 1 Continued on next page*

*Appendix 1—table 1 Continued*

| Family | Pathogen | Antigen | (Predicted) function | Def. of seropositivity is based on | Reference |
|---|---|---|---|---|---|
| Rubella virus* | RV | E1 | Class II viral fusion protein | NA | *Brenner et al., 2019b* |
| *C. trachomatis* | Ct | pGP3 | Virulence factor | NA | *Trabert et al., 2019* |
| *C. tetani** | Ct | TetX | Toxoid (heavy chain) | NA | *Brenner et al., 2019b* |
| *Corynebacterium diphtheriae** | Cd | DTA | Toxoid (intracellular) | NA | *Brenner et al., 2019b* |
| *F. nucleatum* | Fn | Fn0264 | Adhesin (FadA) | At least one positive (experimental) | *Butt et al., 2019* |
| | | Fn1449 | Type Va secretion system (Fap2) | | |
| | | Fn1859 | Porin (FomA) | | |
| *H. pylori* | Hp | HP 10 GroEL | Chaperonin | At least three positive | *Michel et al., 2009* |
| | | HP 73 UreaseA | Urease alpha subunit | | |
| | | HP 547 CagA | Pathogenesis | | |
| | | HP 875 Catalase | Detoxification | | |
| | | HP 887 VacA | Pathogenesis | | |
| | | HP 1564 OMP | Cell envelope | | |
| *S. gallolyticus* | Sg | Gallo2178 | *Pil1 pilus subunit (major pilin)* | NA | *Butt et al., 2016; Butt et al., 2017* |
| *T. gondii* | Tg | p22 | Surface protein | At least one positive | *Brenner et al., 2019a* |
| | | sag-1 | Surface protein | | |

*Pathogens not taken forward due to lack of vaccination history in CoLaus|PsyCoLaus and/or the difficulty in identifying target antigens to ensure specificity of the test.

**Appendix 1—table 2.** Association of risk factors with coronary heart disease (CHD) based on the univariable Cox proportional hazard analyses.

| Variable | HR* (95% CI)* | p |
|---|---|---|
| **Baseline characteristics** | | |
| SCORE2 | 1.72 (1.61–1.85) | $2.87 \times 10^{-61}$ |
| Statin | 3.82 (2.80–5.22) | $3.13 \times 10^{-17}$ |
| Average gross monthly income | 0.85 (0.76–0.96) | $7.27 \times 10^{-3}$ |
| **Genetics** | | |
| CHD-PRS | 1.32 (1.16–1.51) | $4.29 \times 10^{-5}$ |
| PC1 | 74-28 (0.03–195096) | 0.28 |
| PC2 | 0.12 (00.0–728) | 0.64 |
| PC3 | 0.33 (0.00–1131) | 0.79 |
| **Biomarkers of inflammation** | | |
| hs-CRP[†] | 1.91 (1.42–2.55) | $1.51 \times 10^{-5}$ |
| TNF-α[†] | 1.43 (1.05–1.96) | $2.46 \times 10^{-2}$ |
| IL-1β[†] | 0.93 (0.70–1.25) | 0.64 |

*Appendix 1—table 2 Continued on next page*

*Appendix 1—table 2 Continued*

| Variable | HR* (95% CI)* | p |
|---|---|---|
| IL-6[†] | 1.10 (0.88–1.37) | 0.42 |
| **Human polyomaviruses** | | |
| BKPyV | 1.05 (0.72–1.52) | 0.80 |
| JCPyV | 1.14 (0.87–1.50) | 0.35 |
| HPyV6 | 1.66 (1.06–2.61) | 2.74×10-2 |
| WUPyV | 1.45 (0.65–3.27) | 0.37 |
| **Human herpes viruses** | | |
| HSV-1 | 1.88 (1.30–2.68) | 6.52×10-4 |
| HSV-2 | 1.05 (0.74–1.50) | 0.78 |
| CMV | 1.00 (0.76–1.31) | 0.99 |
| EBV | 0.97 (0.46–2.06) | 0.94 |
| HHV-6A | 1.39 (1.03–1.86) | 2.89×10-2 |
| HHV-6B | 0.93 (0.70–1.23) | 0.59 |
| HHV-7 | 0.79 (0.60–1.03) | 8.33×10-2 |
| KSHV | 0.89 (0.44–1.80) | 0.74 |
| VZV | 1.70 (1.02–2.82) | 4.25×10-2 |
| **Parvovirus** | | |
| PVB-19 | 0.90 (0.68–1.21) | 0.49 |
| **Bacteria** | | . |
| *C. trachomatis* | 1.45 (1.11–1.91) | 7.22×10-3 |
| *F. nucleatum* | 1.66 (1.20–2.29) | 2.32×10-3 |
| *H. pylori* | 0.95 (0.67–1.34) | 0.78 |
| *S. gallolyticus* | 0.62 (0.26–1.51) | 0.29 |
| **Parasite** | | |
| *T. gondii* | 1.17 (0.90–1.54) | 0.25 |
| **Pathogen burden** | 1.11 (1.03–1.18) | $3.25×10^{-3}$ |

*HR = hazard ratio, CI = confidence interval.

[†]log10-transformed.

## Appendix 2

### Additional figures

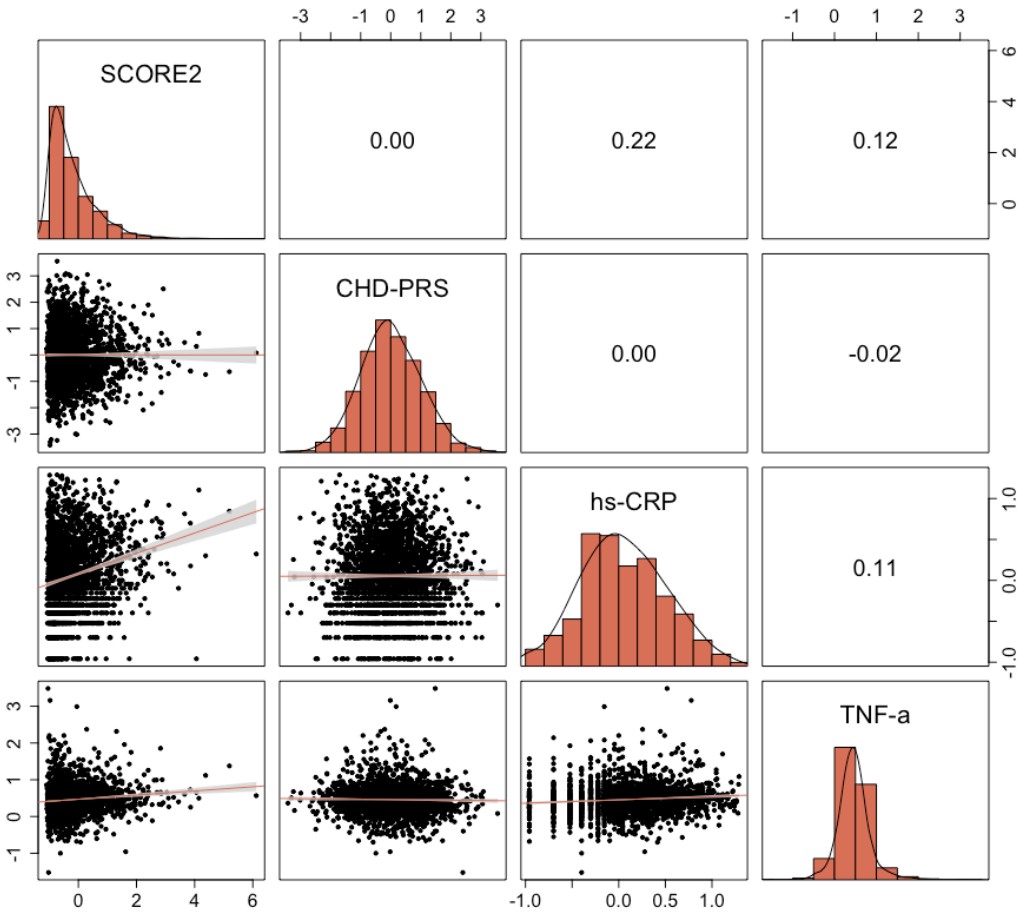

**Appendix 2—figure 1.** Pairwise correlations between quantitative characteristics significantly associated with coronary heart disease (CHD) risk in the univariable Cox proportional hazard models. Pearson's correlation values are displayed, along with linear fits between variables.

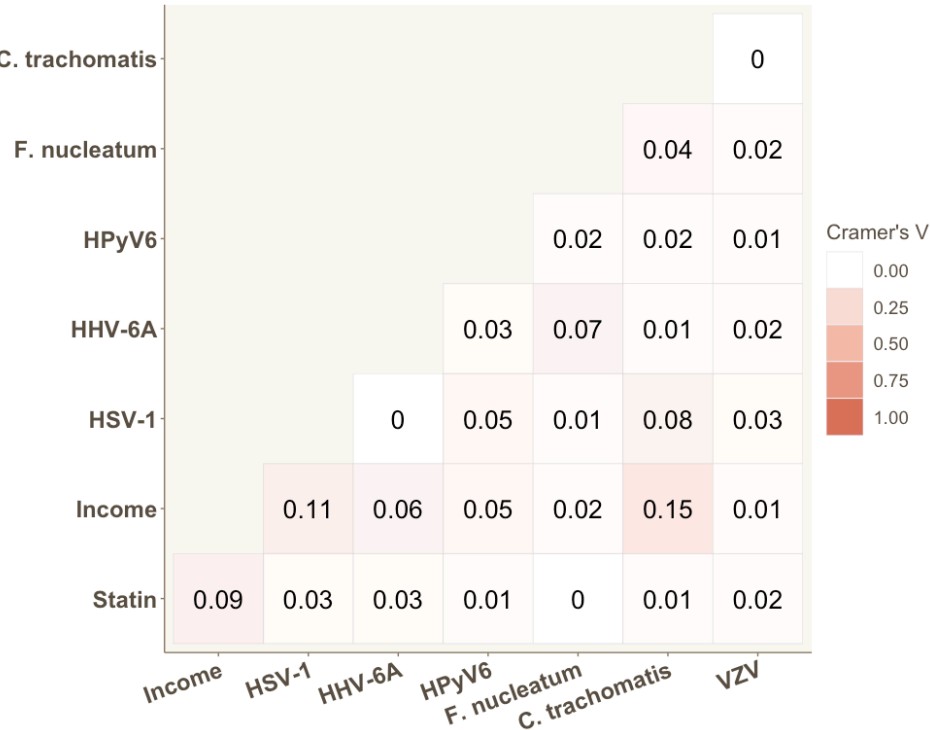

**Appendix 2—figure 2.** Strength of association for each pair of categorical variables significantly associated with coronary heart disease (CHD) risk in univariable Cox proportional hazard models. Cramer's V values are displayed.

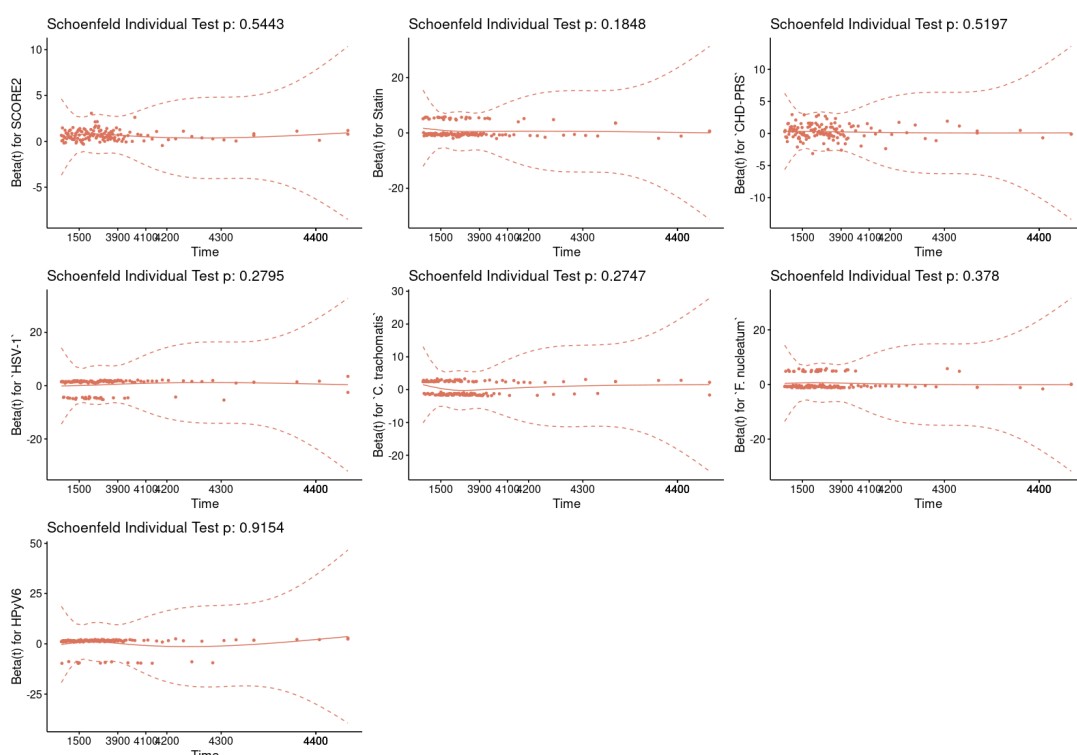

**Appendix 2—figure 3.** Graphical test of proportional hazards assumption (Schoenfeld test). The graphs show the scaled Schoenfeld residuals over time. The p-values (**p**) of the variables and the model as a whole were shown in the plot. A significant p-value (< 0.05) indicates that the variable violates the proportional hazard assumption. The solid line represents the smoothing fitted spline, and the dashed lines the confidence bands at two standard errors. Global Schoenfeld test p=0.58.

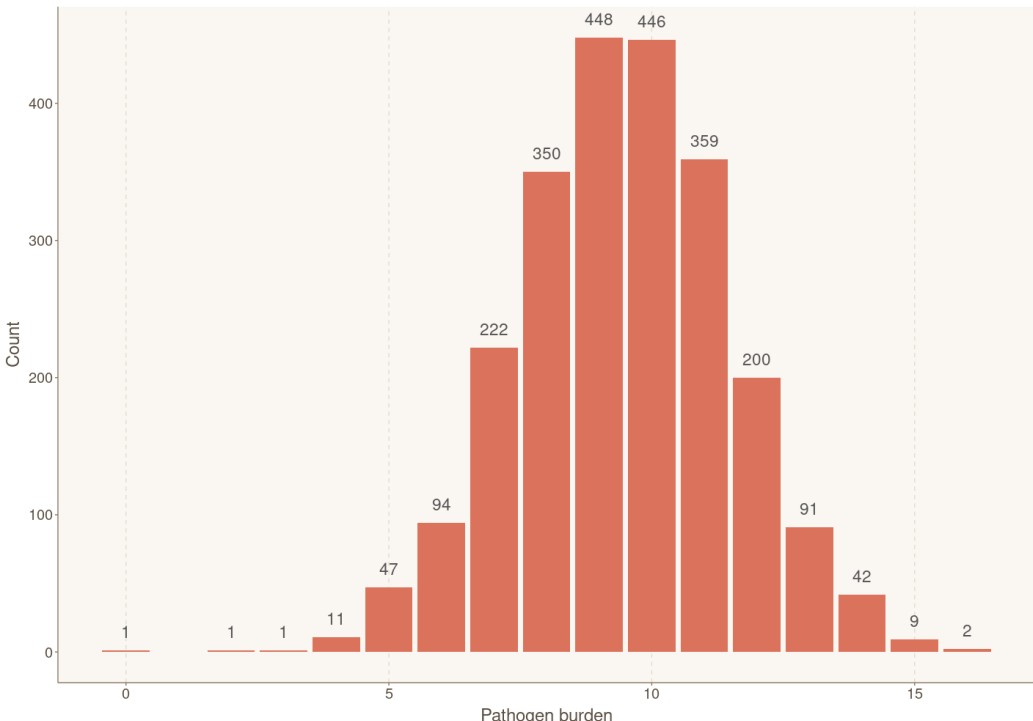

**Appendix 2—figure 4.** Distribution of individuals according to their exposure to infectious agents (pathogen burden). Bar plot showing the number of participants for each cumulative number of positive serological results, reflecting simultaneous ongoing chronic/latent infections. Sample sizes for each group are shown above the box.

