## [Editor Report]

This study is an important contribution to the understanding of cardiovascular disease aetiology based on solid design and methodology. It is a useful independent replication of the effects of traditional risk factors in a large prospective cohort and a valuable investigation of the role of past infection with a commensal bacterium F. nucleatum which warrants validation.

---

## [Decision Letter]

**Decision letter after peer review:**

Thank you for submitting your article "Associations of genetic and infectious risk factors with coronary heart disease" for consideration by *eLife*. Your article has been reviewed by 2 peer reviewers, one of whom is a member of our Board of Reviewing Editors, and the evaluation has been overseen by David James as the Senior Editor. The following individual involved in review of your submission has agreed to reveal their identity: Nabila Bouatia-Naji (Reviewer #2).

Essential revisions:

1. You should limit your description of the traditional risk factors and genetic risk scores as this is not the main focus of their analyses. An introduction to the expected mechanisms linking infection and inflammation and link with atherosclerosis is missing. It would also be good to have a comment on any pre-existing GWAS on seropositivity for F. nucleatum.

2. A brief discussion should be included on the implications of conducting multiple tests

*Reviewer #1 (Recommendations for the authors):*

1. Multiple risk factors were investigated in univariable analysis before their combination into multivariable models. The authors should add to the discussion multiple testing implications

2. Why was no statistical interaction considered?

3. Was there any correlation between Fusobacterium nucleatum with hs-CRP or TNF-a? It would be interesting to know if inflammation is predictive of CHD in multivariable models excluding F. nucleatum

4. Adjusting principal components should be described in Methods not in Results

5. Table 1 title shows n=3462 subjects instead of 3459

6. Table 1 shows Statin=No for 8.6% of the cohort but Figure 2 shows that is probably Statin=yes instead

7. On figure 2 the numbers for all factors do not add up to those in table 1 - please indicate if it is due to missingness in multivariable analysis

8. Supplementary figure 1: scatter plot involving income looks odd. Best fit line does not make sense with income categorised

9. Supplementary figure 2: correlation coefficient between binary variables does not make much sense either. Pairwise association as chi2 test or Cramer V is more appropriate

10. Use correct notation for pvalues (e.g. 2.58e-01=0.26; 2.58e-08=3x10-8)

*Reviewer #2 (Recommendations for the authors):*

The manuscript is very well organized and has several strengths. The methodology is very solid.

1. The authors should limit their description of the traditional risk factors and genetic risk scores as this is not the main focus of their analyses. Overall, the investigation of the genetic score seems artificial to me in the context of the scientific questions of the study.

2. Although I do think the results are sound, the study is overall descriptive and do not provide any support toward a mechanism until we get into the discussion.

3. I understand the article is built as an unbiased approach to test the broad hypothesis of how response to infection may predict the risk for CHD, but some introduction to the expected mechanisms linking infection to low grade inflammation in link to atherosclerosis is missing in the current version of the article, and I would have found it helpful to follow the reasoning of the authors.

4. Also, could the authors comment on potentially pre-exisitng GWAS about infection (or seropositivity for *F. nucleatum*)? Their current numbers do not allow this investigation of course.

---

## [Author Response]

Reviewer #1 (Recommendations for the authors):1. Multiple risk factors were investigated in univariable analysis before their combination into multivariable models. The authors should add to the discussion multiple testing implications

Although multiple univariable testing is a useful statistical tool, it can lead to an increase in the Type I error rate, which means that false positives become more likely when a large number of hypotheses are tested simultaneously. One sentence addressing the implication of multiple testing has been added to the discussion.

2. Why was no statistical interaction considered?

No statistical interactions were considered in our analysis because adding an interaction term to the survival analysis model increases the complexity of the model and limits its interpretability. In addition, adding an interaction term to the model may increase the risk of overfitting, which may lead to incorrect conclusions. Finally, the multivariate survival analysis used in this study still allows us to examine interdependency between multiple variables.

3. Was there any correlation between Fusobacterium nucleatum with hs-CRP or TNF-a? It would be interesting to know if inflammation is predictive of CHD in multivariable models excluding F. nucleatum

We performed a t-test to compare levels of biomarkers of inflammation between seronegative and seropositive individuals. The results showed no statistical difference in hs-CRP or TNF-a levels according to *F. nucleatum* serostatus.

Previous studies have shown that elevated levels of inflammatory markers in the blood, such as hs-CRP, are indeed associated with an increased risk of developing CHD (1, 2). In our study, we also observed significant association between CRP and the occurrence of CHD, independent of *F. nucleatum* serological status. However, this signal was not significant anymore with the addition of other variables such as SCORE2 and statin use.

4. Adjusting principal components should be described in Methods not in Results

We thank you for this suggestion. This part has been moved to Methods.

5. Table 1 title shows n=3462 subjects instead of 3459

This has been corrected, thank you for noticing.

6. Table 1 shows Statin=No for 8.6% of the cohort but Figure 2 shows that is probably Statin=yes instead

This has been corrected, thank you for noticing.

7. On figure 2 the numbers for all factors do not add up to those in table 1 - please indicate if it is due to missingness in multivariable analysis

Thank you for noticing, indeed, the multivariate Cox proportional hazards model was run on data from 2’323 individuals with non-missing data. This has been clarified in the manuscript (see Methods > Multivariable Model).

8. Supplementary figure 1: scatter plot involving income looks odd. Best fit line does not make sense with income categorised

We agree with the reviewer, so we have moved income with the other categorical variables into Appendix 2 - Figure 2.

9. Supplementary figure 2: correlation coefficient between binary variables does not make much sense either. Pairwise association as chi2 test or Cramer V is more appropriate

We thank you for this suggestion. We now use Cramer's V coefficients, which are more appropriate than the correlation coefficient in this context, and adapted the Methods section accordingly.

10. Use correct notation for pvalues (e.g. 2.58e-01=0.26; 2.58e-08=3x10-8)

This has been corrected, thank you.

Reviewer #2 (Recommendations for the authors):The manuscript is very well organized and has several strengths. The methodology is very solid.1. The authors should limit their description of the traditional risk factors and genetic risk scores as this is not the main focus of their analyses. Overall, the investigation of the genetic score seems artificial to me in the context of the scientific questions of the study.

The study aims to delineate the effects of genetic variation, infections, and low-grade inflammation on CHD risk. In particular, the genetic score is a measure of genetic variation that can be used to assess an individual's risk of CHD. It was therefore essential for us to describe it, alongside the traditional risks, and to use it throughout the study to control for other factors that might influence the outcome and to assess their relative importance compared with the other factors studied.

2. Although I do think the results are sound, the study is overall descriptive and do not provide any support toward a mechanism until we get into the discussion.

We agree that our study is essentially descriptive, due to the very nature of the analyses. We used statistical approaches to identify associations between several exposures and coronary artery disease. Additional, functional work is required to demonstrate potential causality, which prevents us from providing mechanistic insights in the Results section. Therefore, we think that it is appropriate to restrict speculations about potential mechanisms to the Discussion.

3. I understand the article is built as an unbiased approach to test the broad hypothesis of how response to infection may predict the risk for CHD, but some introduction to the expected mechanisms linking infection to low grade inflammation in link to atherosclerosis is missing in the current version of the article, and I would have found it helpful to follow the reasoning of the authors.

We thank you for highlighting the importance of introducing the expected mechanisms linking infection to low-grade inflammation, in link to atherosclerosis.

The exact mechanisms linking infection to low-grade inflammation and atherosclerosis are still being studied, though some potential pathways have been proposed. One proposed mechanism involves the production of pro-inflammatory molecules in response to infection (3). These molecules, such as cytokines, can increase the activity of cells involved in atherosclerosis, such as macrophages and smooth muscle cells, leading to the formation of plaques and other changes in the walls of arteries (3). Another mechanism is related to the inflammation at the site of vessel wall. Specifically, it is characterized by the presence of the infectious agents within the atherosclerotic plaques but not within normal blood vessels. The infectious consequences on the atherosclerotic plaque can be accelerated progression or a final complication like thrombosis and plaque rupture (4).

A paragraph explaining these mechanisms has been added to the Introduction section.

4. Also, could the authors comment on potentially pre-exisitng GWAS about infection (or seropositivity for *F. nucleatum*)? Their current numbers do not allow this investigation of course.

To our knowledge, no GWAS on *Fusobacterium nucleatum* has been published to date, neither on humoral immune response (i.e., IgG levels) nor on susceptibility to infection / colonization (i.e., serostatus). This information has been added to the Discussion.

References

(1) Casas, J. P., Shah, T., Hingorani, A. D., Danesh, J., & Pepys, M. B. (2008). C‐reactive protein and coronary heart disease: a critical review. *Journal of internal medicine*, *264*(4), 295-314.

(2) C Reactive Protein Coronary Heart Disease Genetics Collaboration. (2011). Association between C reactive protein and coronary heart disease: mendelian randomisation analysis based on individual participant data. *Bmj*, *342*.

(3) Campbell, L. A., & Rosenfeld, M. E. (2015). Infection and atherosclerosis development. *Archives of medical research*, *46*(5), 339-350.

(4) Pedicino, D., Giglio, A. F., Galiffa, V. A., Cialdella, P., Trotta, F., Graziani, F., & Liuzzo, G. (2013). Infections, immunity and atherosclerosis: pathogenic mechanisms and unsolved questions. *International journal of cardiology*, *166*(3), 572-583.